

**Debris flow initiation characteristics and occurrence probability after extreme rainfalls: case study in the Chenyulan watershed, Taiwan**

**Jinn-Chyi Chen[1], Jiang- Guo Jiang[1], Wen-Shuen Huang[2], Yuan-Fan Tsai[3]**

[1]Department of Environmental and Hazards-Resistant Design, Huafan University, Taipei 22301, Taiwan

[2]Ecological Soil and Water Conservation Research Center, National Cheng Kung University, Tainan 70101, Taiwan

[3]Department of Social and Regional Development, National Taipei University of Education, Taipei 10671, Taiwan

*Correspondence to*: Jinn-Chyi Chen (chenjinnchyi@gmail.com)

**ABSTRACT.** Rainfall and other extreme events often trigger debris flows. This study examines the debris flow initiation characteristics and probability of debris flow occurrence after extreme rainfalls. The Chenyulan watershed, central Taiwan, which has suffered from the Chi-Chi earthquake (CCE) and extreme rainfalls, was selected as a study area. The rainfall index (RI) was used to analyze the return period (T) and characteristics of debris flow occurrence after extreme rainfalls. The characteristics of debris flow occurrence included the variation in critical RI, threshold of RI for debris flow initiation, and recovery period ($t_0$), the time required for the lowered threshold to return to the original threshold. The variations in critical RI after extreme rainfall and $t_0$ associated with RI were presented. The critical RI threshold was reduced in the years following an extreme rainfall event. The reduction in RI as well as $t_o$ were influenced by the RI. Reduced RI values showed an increasing trend over time, and it gradually return to initial RI. The empirical relationship between the probability of debris flow occurrence (P) and corresponding T of the rainfall characteristics for areas affected by extreme rainfalls and affected by the CCE were developed. Finally, a method for determining the P of a rainfall event was proposed based on the relationship between P and T. This method was successfully applied to evaluate the probability of debris flow occurrence after extreme rainfalls.

Keywords: Probability, Debris flow occurrence, Rainfall index, Recovery period, Return period,



## 1. INTRODUCTION

Extreme rainfall or high-intensity rainfall events at regional (Trenberth et al., 2007) and global (Beniston, 2009; Giorgi et al., 2011) scales are increasing because of global warming and the associated increase in water vapor content and energy in the atmosphere. Consequently, in many areas, flash flood and debris-flow hazards are expected to increase in severity driven by severe weather in the form of heavy rains (Kleinen and Petschel-Held, 2007; Beniston et al., 2011).

Extreme events such as extreme rainfall and major earthquakes can cause landslides and debris flows in mountainous watersheds, which generally deposit large amounts of loose debris in gullies and on slopes (Dong et al., 2009; Chen et al., 2012) and increase the volume of loose debris within a watershed. The supply of loose debris has an important role in the occurrence of future debris flows and may change the critical rainfall threshold for the initiation of debris flows during subsequent rainfall events (Jakob et al., 2005). In other words, the critical rainfall threshold for debris flow initiation may differ before and after an extreme rainfall event or major earthquake. Therefore, understanding the variations in rainfall characteristics after extreme events and their influence on debris flow initiation is important for the implementation of debris flow warnings and hazard mitigation.

Previous investigators have studied debris flows following major earthquakes, such as the effects of the Chi-Chi earthquake (CCE) on the characteristics of debris flows in Taiwan (Lin et al., 2003; Liu et al., 2008), the variation in rainfall conditions required to trigger debris flows, and the affected period after the CCE (Chen, 2011), as well as the impact of the Wenchuan earthquake in China on subsequent long-term debris flow activity (Zhang and Zhang, 2017). Chen et al. (2012) studied recent changes in the number of rainfall events related to debris flow occurrence. They found that the number of extreme rainfall events in the Chenyulan watershed showed an increasing trend. Chen et al. (2013) analyzed the characteristics of rainfall related to debris flow occurrence in the Chenyulan watershed to investigate the variation in the rainfall conditions related to debris flow occurrences, and the empirical relationship between rainfall characteristics and the corresponding number of debris flows. Extreme rainfall events and the CCE were shown to affect the critical condition for the occurrence of debris flows in the Chenyulan watershed. The CCE significantly lowered the critical rainfall threshold for debris flow occurrence in the subsequent five years. However, there is a lack of studies quantifying the decrease in the critical rainfall threshold and the period affected by extreme rainfalls. Furthermore, the return period (T), also referred to as the recurrence interval, is an important concept that reflects the long-term hydrological characteristics of an area, which is useful for hydrological or hydraulic design. Therefore, studies are warranted to determine the relationship between the T of rainfall characteristics associated with the probability of debris flow occurrence (P) and apply the relationship between P and T to evaluate probability of debris flow initiation, especially after rainfall and other extreme events.

The Chenyulan watershed in central Taiwan was selected as a study area, because it has experienced both earthquakes (i.e., the CCE) and extreme rainfall events. This study had three main purposes. (1) Investigate the debris flow initiation characteristics after extreme rainfalls. Initiation characteristics include the variation in the rainfall index (RI) threshold for debris flow initiation (i.e., the variation in the critical RI), and the recovery period (t), the time required for the lowered



threshold to return to the original threshold. (2) Develop an empirical relationship between P and T for areas affected by extreme rainfalls and earthquakes. (3) Apply the P–T empirical relationship to evaluate the probability of debris-flow occurrence after extreme rainfalls.

**2. Debris flows in the Chenyulan watershed**

As the study area, the watershed of the Chenyulan River, located in Nantou County, central Taiwan (Figure 1), has an area of 449 km$^2$, main stream length of 42 km, average stream-bed gradient of 4°, and elevation range of 310–3,952 m. The annual regional rainfall in the watershed is between 2,000 and 5,000 mm, with an average of approximately 3,500 mm. Approximately 80% of the annual rainfall occurs in the rainy season between May and October, especially during typhoons,

which generally occur three or four times annually. The CCE occurred on September 21, 1999 with a magnitude of 7.6 on the moment magnitude scale and 7.3 on the Richter scale, and was the largest earthquake in Taiwan in 100 years (Shin and Teng, 2001). It caused numerous landslides in the Chenyulan watershed. Owing to steep topography, loose soils, young (3 million years) and weak geological formations due to ongoing orogenesis, heavy rainfall, and active earthquakes, many debris flows were triggered by more than 30 rainfall events, including five extreme rainfall events, in past five decades in the

watershed (Chen et al., 2013).

**2-1 Rainfall index and extreme rainfall events**

The occurrence of debris flow depends not only on the accumulated rainfall but also on the rainfall intensity, the RI, defined as the product of the maximum 24-h rainfall ($R_d$) and the maximum hourly rainfall ($I_m$, i.e., $RI = R_d I_m$) of a rainfall

event, which was used to indicate either high accumulated rainfall or high rainfall intensity that could trigger debris flows. The RI was used by Chen et al. (2013) to study the characteristics of rainfall triggering of debris flows in the Chenyulan watershed, Taiwan. Long-term rainfall records were obtained from three meteorological stations (Sun Moon Lake, Yushan, and Alisan stations, as shown in Figure 1). These data were used to estimate the regional rainfall characteristics for the whole Chenyulan watershed, via the reciprocal-distance-squared (RDS) method (Chow et al., 1988).

Table 1 lists the debris flow events and related rainfall characteristics for five extreme rainfalls. The five extreme rainfall events included Typhoon Herb (TH) in 1996, Typhoon Toraji (TT) in 2001, Typhoon Mindulle (TMi) in 2004, a heavy rainstorm (HR) in 2006, and Typhoon Morakot (TM) in 2009. Each of these five events caused ten or more debris flows in the watershed and it had the highest RI values of all rainfall events from 1963 to 2016, with RI > 365 cm$^2$/h, as shown in Figure 2.






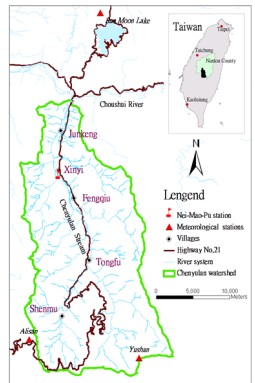

Figure 1: Location of the Chenyulan watershed in central Taiwan.

Table 1: Debris-flow events and related rainfall characteristics for five extreme rainfalls in the Chenyulan watershed (Chen et al., 2013)

| Extreme rainfall event | Date | $I_m$ (mm/h) | $R_d$ (mm) | RI (cm²/h) |
|---|---|---|---|---|
| Typhoon Herb (TH) | July 31–Aug 01, 1996 | 71.6 | 1181.6 | 846.0 |
| Typhoon Toraji (TT) | July 29–30, 2001 | 78.5 | 587.6 | 461.3 |
| Typhoon Mindulle (TMi) | July 02–03, 2004 | 54.0 | 681.4 | 368.0 |
| Heavy rainstorm (HR) | June 08–11, 2006 | 77.5 | 682.8 | 529.2 |
| Typhoon Morakot (TM) | Aug 06–11, 2009 | 85.5 | 1192.6 | 1019.7 |

Note: N = total number of individual debris flows triggered by each rainfall event; $I_m$= maximum hourly rainfall in each rainfall event; $R_d$ = maximum 24-h rainfall amount in each rainfall event; RI = rainfall index, RI = $R_d I_m$

**2-2 Variations in the rainfall index**

Extreme rainfall events and the CCE have been shown to affect the critical conditions required for the occurrence of debris flows, and the critical RI values for occurrence of debris flows have been classified into four categories (Chen et al., 2013), as shown in Figure 2, the periods before TH, between TH and CCE, between CCE and TMi, and between TMi and TM. These periods had critical RI values of approximately 165, 60, 2, and 100 cm²/h, respectively. These trends showed that TH caused numerous landslides and debris flows in the watershed, which reduced the critical rainfall threshold for debris flows in subsequent years and the CCE significantly lowered the critical rainfall threshold for debris flow occurrence in the




subsequent five years. After the CCE, the critical RI dropped sharply to approximately 2 cm²/h, which was 30 times lower than that before the CCE (critical RI = 60 cm²/h). The results also showed that, approximately five years after the CCE, the critical RI gradually recovered from 2 cm²/h to 100 cm²/h (i.e., the critical RI between TMi and TM).

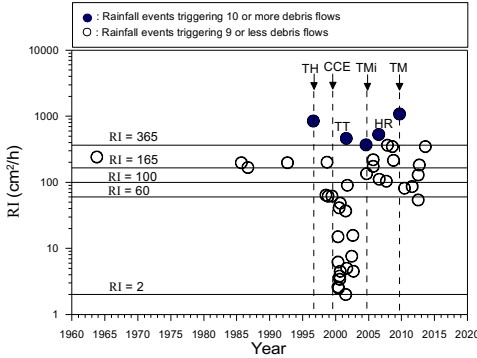

5   Figure 2: The variations in rainfall index (RI) for all rainfall events triggering debris flows between 1963 and 2016 in the Chenyulan watershed. The critical RI for debris flow occurrence in the years following the CCE was much smaller than those before the earthquake and five years after the earthquake. (Modified from Chen et al., 2013)

**3. Variations in the rainfall index after extreme events**

10   The extreme events in the Chenyulan watershed included a severe earthquake, the CCE, and five extreme rainfalls (TH, TT, TMi, HR, and TM). The extreme rainfall events and the severe earthquake affected the critical condition for debris flow occurrence. Here, the variation ratio in RI (i.e., the index $r_R$) was used to evaluate the affected period for the variation of RI after an extreme event. $r_R$ is defined as $RI_t/RI_o$, in which $RI_t$ is the value of the critical RI to trigger debris flow at the t years after an extreme event; $RI_o$ is the critical RI unaffected by extreme events such as extreme rainfalls or the CCE, where $RI_o =$

15   165 cm²/h (i.e., the critical RI before TH), as shown in Figure 2. $r_R < 1.0$ indicates that the critical RI to initiate debris flow is lower than that unaffected by extreme events. $r_R = 1.0$ indicates that the critical RI after an extreme event is equal to that before TH (= 165 cm²/h), and the critical RI has returned to that unaffected by extreme events.

Table 2 shows the estimated period of time t from the time of an extreme event, including TH, CCE, TT, TMi, HR, or TM, and the $r_R$ value between extreme events. Data on the $r_R$ value corresponding with time t between extreme events are plotted





in Figure 3. The empirical curve, determined from the lower bound of the data, at each extreme event could be obtained. Furthermore, two empirical parameters, the initial stage of $r_R$ ($r_{R0}$) and the recovery period ($t_o$), could be obtained from the empirical curve. $t_o$ represents the period for the critical RI affected by an extreme rainfall, which can be determined from an estimation of the required period for $r_R$ to change from the initial value at $r_{R0}$ (< 1) to $r_R$ = 1. For example, one can obtain $r_{R0}$

= 0.8 and $t_o$ = 1.2 years for RI = 368 cm$^2$/h driven by the extreme rainfall of TMi, and $r_{R0}$= 0.65 and $t_o$ = 2.3 years for RI = 529 cm$^2$/h driven by HR. Each curve showed that $r_R$ at the initial stage (t = 0), $r_{Ro}$, was the lowest (with $r_R$ < 0) and $r_R$ gradually increased with increasing t. $r_R$ < 0 indicates that that the critical RI after an extreme event is lower than $RI_o$, and the lowered value depends on the RI value driven by extreme rainfall. Among the empirical curves, the lowest $r_{Ro}$ was caused by the CCE, as shown the dash line in Figure 3. The recovery period $t_o$ affected by CCE was approximately 5 years that agreed

with the results of Chen (2011). The impact of the CCE on the critical RI was more significant than those of the extreme rainfalls.

Figure 4 shows the relationship between $r_{R0}$ and RI and the relationship between $t_o$ and RI. The recovery period ($t_o$) was between 1 and 3 years and depended on RI. $r_{R0}$ decreased and $t_o$ increased with increasing RI, indicating that a rainfall event

with a higher RI resulted in a decrease in the critical RI and longer period affected by the extreme rainfall. Understanding the empirical relationships for $r_{R0}$ and $t_o$ against RI are helpful for modifying the criteria of debris flow warnings after suffering extreme rainfalls in the Chenyulan watershed. For example, $r_{R0}$ = 0.4 and $t_o$ = 3.2 years when an extreme rainfall with RI = 900 cm$^2$/h based on the curves showed in Figure 4. This indicates that the critical RI after an extreme rainfall event could be modified to 40% of the original criteria ($RI_o$ = 165 cm$^2$/h) to RI = 66 cm$^2$/h, and the period of critical RI could be lowered to

approximately three years.



Table 2: Debris flow events and related rainfall characteristics between extreme rainfalls and the Chi-Chi earthquake (CCE) in the Chenyulan watershed between 1996 and 2016

| Year | Date of the event | Name of the event | $I_m$ (mm/h) | $R_d$ (mm) | RI (cm²/h) | t (y) | $r_R$ | Analysis range |
|---|---|---|---|---|---|---|---|---|
| 1996 | July 31–Aug 01 | Typhoon Herb | 71.6 | 1181.6 | 846 | | | TH |
| 1998 | June 07–08 | Rainstorm | 28.1 | 227.8 | 64 | 1.85 | 0.39 | |
| 1998 | Aug 04–05 | Typhoon Otto | 64.6 | 311.7 | 201.4 | 2.01 | 1.22 | (1) |
| 1998 | Oct 15–16 | Typhoon Zeb | 24.6 | 251 | 61.7 | 2.21 | 0.37 | |
| 1999 | May 27–28 | Rainstorm | 24.3 | 254.3 | 61.8 | 2.83 | 0.37 | |
| 1999 | Sep 21 | Chi-Chi earthquake | | | | | | CCE |
| 2000 | Apr 1 | Rainstorm | 20 | 75.1 | 15 | 0.53 | 0.09 | |
| 2000 | Apr 25 | Rainstorm | 8.4 | 30.6 | 2.6 | 0.59 | 0.02 | |
| 2000 | Apr 28–29 | Rainstorm | 7.9 | 78.2 | 6.2 | 0.61 | 0.04 | |
| 2000 | May 2 | Rainstorm | 8.1 | 30.6 | 2.5 | 0.61 | 0.02 | |
| 2000 | June 12–14 | Rainstorm | 18 | 228.1 | 41.1 | 0.73 | 0.25 | |
| 2000 | July 18 | Rainstorm | 12.7 | 30 | 3.8 | 0.82 | 0.02 | (2) |
| 2000 | July 22 | Rainstorm | 16.3 | 20.7 | 3.4 | 0.84 | 0.02 | |
| 2000 | Aug 5 | Rainstorm | 11.6 | 38.8 | 4.5 | 0.87 | 0.03 | |
| 2000 | Aug 22–23 | Typhoon Bilis | 20.6 | 234.5 | 48.3 | 0.92 | 0.29 | |
| 2001 | Jun 5 | Rainstorm | 7.5 | 27 | 2 | 1.71 | 0.01 | |
| 2001 | June 14–15 | Rainstorm | 18.4 | 200.1 | 36.8 | 1.73 | 0.22 | |
| 2001 | July 29–30 | Typhoon Toraji | 78.5 | 587.6 | 461.3 | | | TT |
| 2001 | Aug 10 | Rainstorm | 22.4 | 22.4 | 5 | 0.03 | 0.03 | |
| 2001 | Sep 17 | Typhoon Nari | 35.7 | 252.5 | 90.1 | 0.13 | 0.55 | |
| 2002 | May 31 | Rainstorm | 14.4 | 53 | 7.6 | 0.84 | 0.05 | (3) |
| 2002 | July 03–04 | Rainstorm | 13.3 | 117.9 | 15.7 | 0.93 | 0.10 | |
| 2002 | Aug 12 | Rainstorm | 17.1 | 26.5 | 4.5 | 1.03 | 0.03 | |
| 2004 | July 02–03 | Typhoon Mindulle | 54 | 681.4 | 368 | | | TMi |
| 2004 | Aug 23–25 | Typhoon Aere | 35 | 385.4 | 134.9 | 0.14 | 0.82 | |
| 2005 | Aug 04–05 | Typhoon Matsa | 42.3 | 411.9 | 174.2 | 1.09 | 1.06 | (4) |
| 2005 | Aug 31–Sep 01 | Rainstorm | 44.3 | 495 | 219.3 | 1.16 | 1.33 | |
| 2006 | June 08–11 | Heavy Rainstorm | 77.5 | 682.8 | 529.2 | | | HR |
| 2006 | July 13–15 | Typhoon Bilis | 29.9 | 371.7 | 111.1 | 0.09 | 0.67 | |
| 2007 | Aug 17–20 | Typhoon Sepat | 31.6 | 328.4 | 103.8 | 1.19 | 0.63 | |
| 2007 | Oct 06–07 | Typhoon Krosa | 54.3 | 669.4 | 363.5 | 1.32 | 2.20 | (5) |
| 2008 | July 17–18 | Typhoon Kalmaegi | 67.2 | 515.7 | 346.6 | 2.10 | 2.10 | |
| 2008 | Sep 12–15 | Typhoon Sinlaku | 35 | 612.4 | 214.3 | 2.26 | 1.30 | |
| 2009 | Aug 06–11 | Typhoon Morakot | 85.5 | 1192.6 | 1019.7 | | | TM |
| 2010 | May 23–24 | Rainstorm | 35.8 | 227.2 | 81.3 | 0.78 | 0.49 | |
| 2011 | July 17–20 | Rainstorm | 33.6 | 256.2 | 86.1 | 1.94 | 0.52 | |




| 2011 | Aug 28–31 | Typhoon Nanmadol | 8.6 | 93.5 | 8 | 2.06 | 0.05 | |
| 2012 | June 9–12 | Rainstorm | 33.6 | 384.6 | 129.2 | 2.84 | 0.78 | (6) |
| 2012 | June 18–21 | Typhoon Talim | 22.2 | 243.4 | 54 | 2.86 | 0.33 | |
| 2012 | Aug 1–3 | Typhoon Saola | 36.4 | 502.2 | 182.8 | 2.98 | 1.11 | |
| 2013 | July 12–13 | Typhoon Soulik | 52.4 | 661.7 | 346.7 | 3.92 | 2.10 | |
| 2013 | Aug 21–23 | Typhoon Trami | 40.8 | 455.3 | 185.8 | 4.03 | 1.13 | |
| ~2016 | No debris flow event | | | | | | | |

Notes: $I_m$ = maximum hourly rainfall during each rainfall event; $R_d$ = maximum 24-h rainfall amount during each rainfall event; RI = rainfall index, RI = $R_d$ $I_m$; t = the estimated period from the time of an extreme event; $r_R$ = the variation ratio in RI, $r_R$ = $RI_t/RI_o$, where $RI_t$ is the value of critical RI required to trigger debris flow at t years after an extreme event; $RI_o$ = the critical RI unaffected by extreme events such as extreme rainfalls or the CCE, $RI_o$ = 165 cm²/h (the critical RI before TH); (1) = data between TH and CCE; (2) = data between CCE and TT; (3) = data between TT and TMi; (4) = data between TMi and 5   HR; (5) = data between HR and TM; (6) = data after TM.

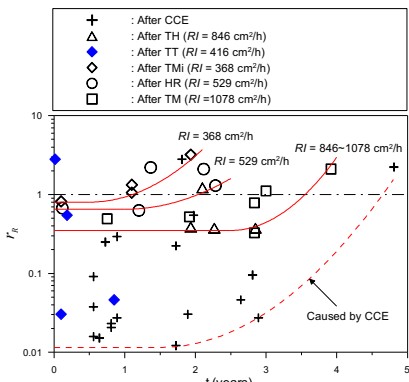

Figure 3: Relationship between the variation ratio in RI ($r_R$) at the initial stage ($r_{R0}$) and the period after an extreme rainfall (t) 10   for the RI driven by an extreme rainfall event




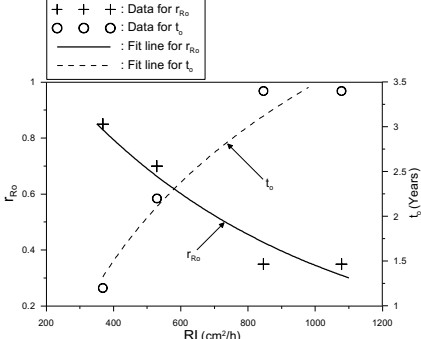

Figure 4: Relationship between the variation ratio in RI ($r_R$) at initial stage ($r_{R0}$) and the RI and the relationship between the recovery period ($t_o$) and RI

## 4. Relationship between the probability of debris flow occurrence and return period

Return period T is the average interval in years between events equaling or exceeding a certain magnitude. T responds the long-term hydrological characteristics of an area and is useful for hydrological or hydraulic design. Therefore, the RI

10    associated with T was determined, and the relationship between the probability of debris flow occurrence and T was developed.

### 4-1 Return period of rainfall

Many methods, such as the formulas by Weibull, Jenkinson, and Gringorten, the computational methods, as well as the modified Gumbel method, have been used to evaluate the return period T of rainfall (Makkonen, 2006). The Weibull

15    formula was used to estimate T for the rainfall of annual maximum series in this study because it can predict much shorter return periods of extreme events than the other methods (Makkonen, 2006). T can be estimated by the Weibull formula as

$$T = (n+1) / m \qquad (1)$$





Where n refers to the number of years in the record and m is the rank of a value in a list ordered by descending magnitude. The RI data of the annual maximum series collected in the Chenyulan watershed between 1960 and 2016 were used to determine T. Figure 5 shows the relationship between RI and T, which can be expressed as

5      $RI = 180 \, (T-0.98)^{0.44}$                                                     (2a)

or

$T = (RI /180)^{2.27} + b$                                                          (2b)

Eq. (2a) or (2b) provide good estimates of RI with T values of less than 50 years. Debris flows could be triggered at lower RI values, corresponding with lower T values for rainfall events within five years after the CCE, as shown by the cross symbols in Figure 5. Five extreme rainfall events, TMi, TT, HR, TH, and TM, are also shown in Figure 5. Excluding TMi and TT, affected by the CCE (within five years after the CCE), the extreme events mostly
15   had T values exceeding 10 years. The T value for the critical RI affected by the CCE was approximately 1 years, much smaller than that affected by extreme events.

Figure 5: The relationship between the rainfall index (RI) and the return period (T) in the Chenyulan watershed.
20





### 4-2 Probability of debris flow occurrence

The P of debris flow occurrence for a rainfall event greater than a value of RI can be calculated by the number ($N_D$) of rainfall events that have triggered debris flows divided by the number ($N_R$) of rainfall events, where $P = N_D/N_R$. Hence, the P

for a rainfall event with given RI or T values corresponding to RI (based on the data in Figure 5) could be determined. Four empirical curves of P versus RI or T based on different periods were developed. The four periods were: (i) the CCE-affected period (CCEAP), (ii) the extreme rainfall-affected period (ERAP), (iii) the whole period (WP) between 1985 and 2016, and (iv) the WP excluding CCEAP and ERAP. After the CCE, the critical RI dropped sharply to approximately 2 $cm^2$/h, which was 30 times lower than that before the CCE (critical RI = 60 $cm^2$/h) (Figure 2). The CCE significantly lowered the critical

rainfall threshold for debris flow occurrence in the subsequent five years (Chen, 2011; Chen et al., 2013). Hence, the CCEAP was considered to be the period of five years after the CCE. Because the critical RI of debris flow occurrence was affected by extreme rainfalls and the affected periods could be reach three years (Figure 4), data within three years after extreme rainfalls were selected to develop the relationship of P versus RI or T for the ERAP. Finally, the WP considered the data of whole period (1985–2016), including CCEAP and ERAP.

Figure 6 shows the relationships between P and T for the four periods, which can be expressed in the form of a logistic function (Hosmer and Lemeshow, 2000)

$$P = \frac{1}{1 + e^{-(\alpha + \beta T)}}$$

(3)

Where $\alpha$ and $\beta$ are empirical coefficients that can be determined by fitting the given data. The empirical coefficients $\alpha$ and $\beta$ at for the four periods are listed in Table 3.


Table 3: Empirical coefficients of $\alpha$ and $\beta$ at in the four studied periods

| Period | $\alpha$ | $\beta$ |
|---|---|---|
| I. Whole period (WP) (1985–2016) | -1.11 | 0.98 |
| II. Chi-Chi earthquake-affected period (CCEAP) | -50.0 | 48.5 |
| III. Extreme rainfall-affected period (ERAP) | -3.97 | 3.89 |
| IV. WP excluding CCEAP and ERAP | -0.95 | 0.78 |

The fitted curve of the P–T relationship for WP was similar to that of WP excluding CCEAP and ERAP. Because WP used long-term data between 1985 and 2016, excluding the short-term data of CCEAP and ERAP to develop the P–T relationship,

there is no obvious difference. Meanwhile, P rose significantly after an extreme rainfall event or the CCE at the same T or under the same rainfall condition. In particular, the P value affected by CCE was markedly higher than that affected by extreme rainfalls. The benefits of developing the P–T relationship (Figure 5) include that P values can be evaluated at





various T values (or different rainfall conditions) to understand how P is affected by the CCE or extreme rainfall. For example, P = 59% at T = 1.5yr (see black curve in Figure 6), while P increases to 87% after an extreme rainfall event (see blue curve) and P = 100% after the CCE (see red curve).

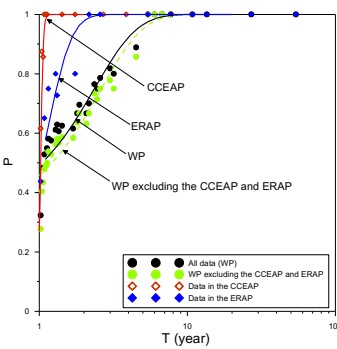

Figure 6: The P–T relationship for the Chi-Chi earthquake-affected period (CCEAP), extreme rainfall-affected period (ERAP), the whole period (WP), and the WP excluding the CCEAP and ERAP

## 5. Application of the empirical model

The heavy rainfall brought by TM in August 2009 had a maximum hourly rainfall of 123 mm and 48-h rainfall of 2,361 mm (measured at Alishan rainfall station), which caused numerous debris flows that buried more than 20 houses in Shenmu, Tongfu, and Xinyi villages (Chen et al., 2011) in the Chenyulan watershed. The relationship between P and T were applied to evaluate the probability of debris flow occurrence after recent rainfall events, such as TM.

The P–T relationship of the WP is

$$P = \frac{1}{1 + e^{1.11 - 0.98T}} \qquad (4)$$

and the P–T relationship of the ERAP is

$$P = \frac{1}{1 + e^{3.97 - 3.89T}} \qquad (5)$$





where T is associated with RI and can be expressed by Eq. (2), i.e., $T = (RI/180)^{2.27} + 0.98$. The probability of debris flow occurrence can be determined when RI is given according to Eq. (4) or Eq. (5).

However, the two equations were developed based on different periods and different data sets, and the valid conditions for the two equations may not be identical. Eq. (4) predominantly reflects the long-term characteristics of debris flow occurrence, and cannot reflect the short-term characteristic caused by extreme events. In contrast, Eq. (5) focuses on the influence of extreme rainfall events. Hence, field data of debris flow occurrence and rainfall between 2012 and 2014 were collected to assess the proposed equations.

Figure 7 shows the variation in the predicted P (blue line) by Eq. (4) from 2012 to 2014, and the peaks labeled with "OCC" in the figure represent debris flow events. There were five debris flow events (Table 4) and most data, four of five, were reasonably predicted by Eq. (4), that four predicted P are exceeding 50%. One debris flow event during Typhoon Talim was not predicted successfully in association with the events occurring within three years after the extreme rainfall event TM. The RI for debris flow occurrence decreased in the early stage after the extreme rainfall event owing to the fact that extreme rainfalls result in large amounts of loose debris in gullies and on slopes. When the P–T relationship in ERAP (Eq. (5)) was used, instead of Eq. (4), the predicted P (red line) with P > 50% was in agreement with the field data of debris flow occurrence.

Table 4: Five debris flow events in the Chenyulan watershed after the extreme rainfall event of Typhoon Morakot between 2012 and 2014

| Year | Date | Rainfall event | $I_m$ (mm/h) | $R_d$ (mm) | RI (cm$^2$/h) |
|------|------|----------------|----------|----------|-------------|
| 2012 | June 9–12 | Rainstorm | 33.6 | 384.6 | 129.2 |
| 2012 | June 18–21 | Typhoon Talim | 22.2 | 243.4 | 54 |
| 2012 | Aug 1–3 | Typhoon Saola | 36.4 | 502.2 | 182.8 |
| 2013 | July 12–13 | Typhoon Soulik | 52.4 | 661.7 | 346.7 |
| 2013 | Aug 21–23 | Typhoon Trami | 40.8 | 455.3 | 185.8 |





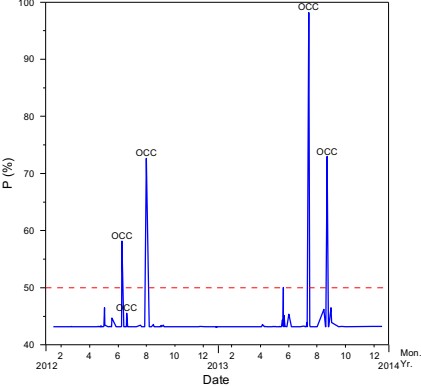

Figure 7: Application of the probabilistic model of debris flow occurrence in the whole period in the Chenyulan watershed between 2012 and 2014.

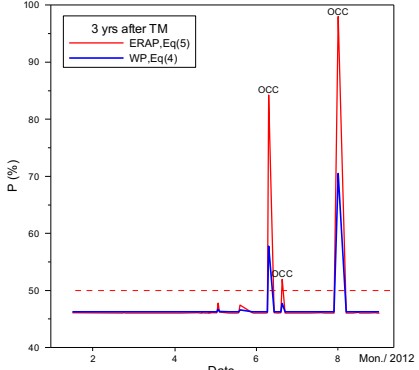

Figure 8: Probabilistic model of debris flow occurrence in the whole period (WP) compared with that in the rainfall-affected period (ERAP)





### 6. Conclusions

Debris flows and their corresponding rainfall events were studied in the Chenyulan watershed, central Taiwan, between
1985 and 2016. The rainfall index RI, defined as the product of $I_m$ and $R_d$, was used to analyze the rainfall conditions critical
for debris flow occurrence after extreme events. The extreme events included the CCE in 1999 and five extreme rainfalls in
1996, 2001, 2004, 2006, and 2009.

The extreme rainfall events and the CCE affected the critical condition for the occurrence of debris flows. RI could reflect
the debris flow initiation characteristics after extreme rainfalls. The critical RI threshold for the occurrence of debris flows
was reduced in the years following an extreme rainfall event. Reduced RI values showed an increasing trend over time, and
it gradually return to initial RI, representative of the RI unaffected by the extreme rainfall. The required time, i.e., the
recovery period ($t_o$), for the decreased RI to increase to the original value for extreme rainfalls was analyzed. The reduction
in RI as well as $t_o$ were influenced by the RI. The RI at the early stage after an extreme rainfall showed the maximum
decrease of approximately 30% of the original RI. The maximum $t_o$ was approximately three years. Understanding the
reduced RI and $t_o$ are helpful for modifying the criteria of debris flow warnings after suffering extreme rainfalls in the
Chenyulan watershed.

The rainfall index (RI) associated with return period (T) was analyzed. The extreme events triggering numerous debris
flows, excluding events affected by CCE, mostly had T values exceeding 10 years. The T value for the critical RI affected by
the CCE was approximately 1 year, much smaller than that affected by extreme events. The empirical relationships between
the probability of debris flow occurrence P and T based on the CCE and extreme rainfalls were developed. P increased
significantly after extreme rainfall events or the CCE at the same T. In particular, the P value influenced by the CCE was
markedly higher than that affected by the extreme rainfall. The relationship between P and T was applied to evaluate P
during recent rainfall events after the extreme rainfall of TM, which showed that the model was reasonable for explaining
debris flow occurrence. The benefits of developing the P–T relationship include that P values can be evaluated at various T
values (or different rainfall conditions) to understand how P is affected by the CCE or extreme rainfall.

### Acknowledgement
This study received financial support from the Ministry of Science and Technology, Taiwan (MOST 105-2625-M-211-001).





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
