# Peer review of "Debris flow initiation characteristics and occurrence probability after extreme rainfalls: case study in the Chenyulan watershed, Taiwan"

_Natural Hazards and Earth System Sciences, 2017_

## Referee Comment (RC1) · Anonymous Referee #1 · 26 Aug 2017

The manuscript presents a statistical analysis of debris flow occurrence after extreme rainfall. The rainfall index was used to analyse the return period and the characteristic of debris flow occurrence after extreme rainfalls in an area also affected by earthquakes. An empirical method based on a relationship between probability of debris flow occurrence and return period was developed and it is successfully applied to the case of Chenyulan watershed (Taiwan). The main purposes of the manuscript are clearly defined. Even though the scientific question, the method is interesting and the scientific approaches are valid, the authors should modify the structure of the manuscript presenting the method before the case study. The desirable outline of manuscript should be : Introduction; Model and related sub-paragraph about model input data, variables

and outputs; Case study and results of the model; conclusion. This is basically a good paper. The authors however need to address the following comments before being accepted: - Debris fall initiation should be replaced by debris fall triggering (e.g. lines 17 and 22 p.1); - Line 19 p.3 the equation should be labelled . The number of RI euation could be cited in Table 1. - Figure 1 should be improved (blue instead of light blue). - More details on study area and characteristic of debris flow (as volume involved and run out area and kind of soil involved) should be added. - The applicability or unapplicability of the model to other case study could be further discussed in the section 6 (Conclusion) - Numbering of sub-paragraph is suggested.

––––––––––––––––––––––––––

---

## Referee Comment (RC2) · Anonymous Referee #2 · 5 Sep 2017

**General comments**

The paper deals with an empirical/statistical analysis of debris flow occurrence base on a variable the authors introduce, the rainfall index RI, defined as the product between the maximum 24-h rainfall and the maximum hourly rainfall of a rainfall event. It has a specific focus on changes of debris flow probability due to the occurrence of previous events (landslide reactivation); so, this makes the MS innovative at some level. The manuscript certainly fits within the scope of the NHESS journal. The methods are overall valid, though the rationale underlying RI is not that clear and supported (see specific comments below). The MS requires an overall revision of the language, as an

too much of symbols and acronyms are used, which makes it very difficult to follow. Finally, I suggest **major revisions** for the manuscript.

**Specific comments**

P2 L 3-5 These are quite strong statements on climate change impacts. Are the authors really sure that the cited papers allow to make these statements?

P2 L 9 ["... and increase the volume of loose debris within a watershed"] The paper analysis is based on the assumption that after an extreme event causing landslides, the probability of landslides increases as a feedback effect. I suppose that in other cases, the opposite may be observed, as the occurrence of landslides can bring to a stabilization of affected slopes, and thus a lower probability of subsequent initiations. The authors should discuss better this issue.

P2 L 28 ["...hydraulic design."] Authors should here add some literature on previous studies focused on the assessment of debris flow/landslide triggering return period. For instance:

*M. Borga, G. Dalla Fontana, F. Cazorzi, Analysis of topographic and climatic control on rainfall-triggered shallow landsliding using a quasi-dynamic wetness index J. Hydrol., 268 (1–4) (2002), pp. 56-71*

*D.J. Peres, A. Cancelliere, Estimating return period of landslide triggering by Monte Carlo simulation, Journal of Hydrology, Volume 541, 2016, Pages 256-271, ISSN 0022-1694, http://dx.doi.org/10.1016/j.jhydrol.2016.03.036.*

*P. D'Odorico, S. Fagherazzi, R. Rigon, Potential for landsliding: dependence on hyeto-graph characteristics J. Geophys. Res.: Earth Surf., 110 (F1) (2005)*

*L. Schilirò, C. Esposito, G. Scarascia Mugnozza, Evaluation of shallow landslide-triggering scenarios through a physically based approach: an example of application in the southern Messina area (northeastern Sicily, Italy), Nat. Hazards Earth Syst. Sci., 15 (9) (2015), pp. 2091-2109*

*Bogaard, T. and Greco, R.: Invited perspectives. A hydrological look to precipitation intensity duration thresholds for landslide initiation: proposing hydro-meteorological thresholds, Nat. Hazards Earth Syst. Sci. Discuss., https://doi.org/10.5194/nhess-2017-241, in review, 2017.*

Section 2. The data section lacks of some essential information: what kind of rainfall data was available (a continuous series? Hourly? Sub-hourly?), how was the rain gauge selected to compute the RI (the "nearest" rain gauge?)

P3 L18-20 the computation of the RI requires a criteria for identifying what is a "rainfall event". The authors should specify the criterion that they have adopted to single-out rainfall events from a rainfall sequence.

Figure 2. In this figure it seems that an "ad hoc" criterion has been used to plot the RI corresponding to events ("10 or more debris flows"). Since it does not seem that the RI has a physically-based/hydrological rationale, the authors should at least better prove if the RI works well in separating triggering and non triggering events. So: what happens if the "10 debris flow" threshold changes (e.g. to 5, or another number)? What happens if the RI index values for NON-triggering events are plotted?

Figure 3. It is unclear to which data points the curves are fitted (or where the curves come from)

Figure 4. It is unclear how this curve has been determined

P1 L18-20; P 6 L4-8; P6 13-15 : (not exhaustive) list of sentences difficult to follow because an excessive use of symbols and acronyms. Write more in terms of "concepts" rather than in terms of "symbols". Perhaps the authors should rewrite the MS with the support of a native-english writer

P 10 L 1: $n$ is the number of years only if one value per year is in the sample (e.g. annual maxima data). From table 2 it seems that multiple values of RI can be present within a year. Please clarify

**Technical corrections**

P2 L 28 warranted is not appropriate. Perhaps use "needed"

P3 L28 replace "it had" with "they had"

---

## Author Comment (AC1) · 3 Nov 2017

Please see the attached file.

Please also note the supplement to this comment:
https://www.nat-hazards-earth-syst-sci-discuss.net/nhess-2017-265/nhess-2017-265-AC1-supplement.pdf
* * *

---

## Author Comment (AC2) · 3 Nov 2017

nhess-2017-265,

**Debris flow initiation characteristics and occurrence probability after extreme rainfalls: case study in the Chenyulan watershed, Taiwan**

Jinn-Chyi Chen, Jiang- Guo Jiang, Wen-Shun Huang, Yuan-Fan Tsai

First Contact: Jinn-Chyi Chen, jinnchyi@cc.hfu.edu.tw

**Response to Reviewer 2**

**I. General comments**

*The paper deals with an empirical/statistical analysis of debris flow occurrence base on a variable the authors introduce, the rainfall index RI, defined as the product between the maximum 24-h rainfall and the maximum hourly rainfall of a rainfall event. It has a specific focus on changes of debris flow probability due to the occurrence of previous events (landslide reactivation); so, this makes the MS innovative at some level. The manuscript certainly fits within the scope of the NHESS journal. The methods are overall valid, though the rationale underlying RI is not that clear and supported (see specific comments below). The MS requires an overall revision of the language, as too much of symbols and acronyms are used, which makes it very difficult to follow. Finally, I suggest major revisions for the manuscript.*

Response:

We would like to thank the reviewer for their detailed comments and suggestions for the manuscript. We believe that the comments have identified important areas that required improvement. Major revisions have been implemented in the manuscript following the reviewer's suggestions.

(1) We have added sections (3-1~3-3) and Figures (2 a and b) to describe, analyze, and explain for the RI index, including the reason for using the index in this study and the rationale underlying RI.

(2) The overall language of the manuscript has been revised. The usage of symbols and acronyms have been reduced and a list of symbols and abbreviations has been added in the Appendix to ensure easy understanding.

(3) We have also responded to the specific comments and technical corrections suggested by the reviewer; point by point responses are as follows.

**II. Specific comments**

*1. P2 L 3-5 These are quite strong statements on climate change impacts. Are the authors really sure that the cited papers allow to make these statements?*

Response: This description in the original manuscript has been deleted.

*2. P2 L 9 [": : : and increase the volume of loose debris within a watershed"] The paper analysis is based on the assumption that after an extreme event causing landslides, the probability of landslides increases as a feedback effect. I suppose that in other cases, the*

*opposite may be observed, as the occurrence of landslides can bring to a stabilization of affected slopes, and thus a lower probability of subsequent initiations. The authors should discuss better this issue.*

Response:

(1) This description in the original manuscript has been revised and the opposite feedback effect that may be caused by landslides has been added (Lines 7-9, Page 2).

(2) The supply of loose debris caused by landslides plays an important role in the occurrence of future debris flows and may change the critical rainfall threshold for the initiation of debris flows during subsequent rainfall events. The related phenomena are presented in Figure 3 and shown in the text, (blue text in Lines 13-28, Page 11).

*3. P2 L 28 [": : :hydraulic design."] Authors should here add some literature on previous studies focused on the assessment of debris flow/landslide triggering return period. For instance:*

*M. Borga, G. Dalla Fontana, F. Cazorzi, Analysis of topographic and climatic control on rainfall-triggered shallow landsliding using a quasi-dynamic wetness index J. Hydrol., 268 (1–4) (2002), pp. 56-71.*

*D.J. Peres, A. Cancelliere, Estimating return period of landslide triggering by Monte Carlo simulation, Journal of Hydrology, Volume 541, 2016, Pages 256-271, ISSN 0022-1694.*

*P. D'Odorico, S. Fagherazzi, R. Rigon, Potential for landsliding: dependence on hyetograph characteristics J. Geophys. Res.: Earth Surf., 110 (F1) (2005).*

*L. Schilirò, C. Esposito, G. Scarascia Mugnozza, Evaluation of shallow landslide triggering scenarios through a physically based approach: an example of application in the southern Messina area (northeastern Sicily, Italy), Nat. Hazards Earth Syst. Sci., 15 (9) (2015), pp. 2091-2109.*

*Bogaard, T. and Greco, R.: Invited perspectives. A hydrological look to precipitation intensity duration thresholds for landslide initiation: proposing hydro-meteorological thresholds, Nat. Hazards Earth Syst. Sci. Discuss., in review, 2017.*

Response: Thank you for the suggestion. We have added the suggested references (blue text in Lines 3-5, Page 3).

*4. Section 2. The data section lacks of some essential information: what kind of rainfall data was available (a continuous series? Hourly? Sub-hourly?), how was the rain gauge selected to compute the RI (the "nearest" rain gauge?).*

Response:

We apologize for the missing information. Essential information associated with rainfall estimation has been added in section 3.

There are only three meteorological stations (Sun Moon Lake, Yushan, and Alisan stations) near/within the Chenyulan stream watershed, as shown in Fig. 1(a), and these stations provide long-term (more than 43 y*r*) records of hourly rainfall data

series. Thus, data from the three meteorological stations were used to estimate the regional rainfall characteristics for the whole Chenyulan watershed by the reciprocal-distance-squared method. The use of the reciprocal-distance-squared method and its limitation are presented and the estimation of the regional rainfall characteristics is expressed as Eq. (1).

*5. P3 L18-20 the computation of the RI requires a criteria for identifying what is a "rainfall event". The authors should specify the criterion that they have adopted to single-out rainfall events from a rainfall sequence.*

Response: The criteria for identifying rainfall event has been added in the section of 3-2, as shown in Lines 10-15, Page 6.

*6. Figure 2. In this figure it seems that an "ad hoc" criterion has been used to plot the RI corresponding to events ("10 or more debris flows"). Since it does not seem that the RI has a physically-based/hydrological rationale, the authors should at least better prove if the RI works well in separating triggering and non triggering events. So: what happens if the "10 debris flow" threshold changes (e.g. to 5, or another number)? What happens if the RI index values for NON-triggering events are plotted?*

Response:

We have added more descriptions is the revised manuscript describing the reasons for using the RI index (see Section 3).

(1) The use of other rainfall indexes is analyzed, as shown in Figure 2 (a) and (b), and the use of rainfall index is discussed. From the analysis of RI index, the index more suitably reflects the critical rainfall to trigger debris flows, especially for extreme rainfall that induced both high rainfall intensity and high accumulated rainfall in the study area.

(2) The five extreme rainfall events discussed in this paper have both characteristics of having high critical RI value and triggering a large number of debris flows (N>10). Most rainfall events, occupying 87%, caused four or less debris flows (N<4). The number of debris flows for the five extreme rainfall events varies significantly from other rainfall events. Thus, we used the rainfall index to separate the two groups of extreme rainfall events (N>10) and non-extreme rainfall events (exactly is N<4, not N<10 in the original manuscript). This description has also been added in the revised manuscript.

(3) It is important to understand how the non-triggering events affect the rainfall index. The analysis that only focused on non-triggering events using the RI index was not presented in this paper because the RI index is developed on the basis of the criteria of debris-flow occurrence, and there are too many non-triggering events, making it difficult to clearly present them. Perhaps the combination of the probability concept to discuss or analyze the effect of non-triggering events is more meaningful. Therefore, we developed the probability model of debris flow occurrence in this study.

*7. Figure 3. It is unclear to which data points the curves are fitted (or where the curves come from).*

Response: The section in the revised manuscript describing Figure 3 has been improved and rewritten, particularly for clarifying the fitting of data points and origin of the curves (Lines 16-27, Page 10, and Lines 1-2, Page 11).

*8. Figure 4. It is unclear how this curve has been determined.*

Response:

The section in the revised manuscript describing Figure 4 has been improved and rewritten.

There are four data sets, either for minimum $r_{RI}$ or for recovery period, in Figure 4. Values of these data are obtained from the critical lines in Figure 3. The method of determining the critical lines has also been described in Lines 2-8, Page 12.

*9. P1 L18-20; P 6 L4-8; P6 13-15 : (not exhaustive) list of sentences difficult to follow because an excessive use of symbols and acronyms. Write more in terms of "concepts" rather than in terms of "symbols". Perhaps the authors should rewrite the MS with the support of a native-english writer.*

Response:

(1) The manuscript has been revised following the reviewer's suggestion and rewritten more in terms of concepts to explain the results of study. The usage of symbols has been reduced, such as $t_0$, $r_{R0}$ and OCC.

An Appendix: List of symbols and abbreviations has also been added to ensure ease of understanding.

(2) We have revised the manuscript with the help of a native-English writer.

*10. P 10 L 1: n is the number of years only if one value per year is in the sample (e.g. annual maxima data). From table 2 it seems that multiple values of RI can be present within a year. Please clarify.*

Response:

The use of n was aimed to evaluate the return period T. In this study, the RI data of the annual maximum series are ranked and collected between 1960 and 2016. The data of the annual maximum RI were used to determine the return period T of rainfall. T responds the long-term hydrological characteristics of an area and is useful for hydrological or hydraulic design.

Table 2 lists numerous debris-flow events triggered by rainstorms and typhoons

between 1996 and 2016, and shows the events of debris flow for their corresponding RI. Because many debris flow events occurred within a year, multiple values of RI can be presented within a year.

For hydrological design, a rainfall event or a value of RI, has a corresponding T in the Chenyulan watershed. The abovementioned explanations are emphasized in the revised manuscript.

**III. Technical corrections**

*1. P2 L 28 warranted is not appropriate. Perhaps use "needed"*

Response: "warranted" has been replaced with "needed" (Line 8, Page 3).

*2. P3 L28 replace "it had" with "they had"*

Response: "it had" has been replaced with "they had" (Line 16, Page 7).

---

## Author Comment (AC3) · 3 Nov 2017

nhess-2017-265,

**Debris flow initiation characteristics and occurrence probability after extreme rainfalls: case study in the Chenyulan watershed, Taiwan**

Jinn-Chyi Chen, Jiang- Guo Jiang, Wen-Shun Huang, Yuan-Fan Tsai

First Contact: Jinn-Chyi Chen, jinnchyi@cc.hfu.edu.tw

**Comments for Reviewer 1**

*The manuscript presents a statistical analysis of debris flow occurrence after extreme rainfall. The rainfall index was used to analyze the return period and the characteristic of debris flow occurrence after extreme rainfalls in an area also affected by earthquakes. An empirical method based on a relationship between probability of debris flow occurrence and return period was developed and it is successfully applied to the case of Chenyulan watershed (Taiwan).*

*The main purposes of the manuscript are clearly defined. Even though the scientific question, the method is interesting and the scientific approaches are valid, the authors should modify the structure of the manuscript presenting the method before the case study. The desirable outline of manuscript should be Introduction; Model and related sub-paragraph about model input data, variables and outputs; Case study and results of the model; conclusion.*

*This is basically a good paper. The authors however need to address the following comments before being accepted: - Debris fall initiation should be replaced by debris fall triggering (e.g. lines 17 and 22 p.1); - Line 19 p.3 the equation should be labelled. The number of RI equation could be cited in Table 1. - Figure 1 should be improved (blue instead of light blue). - More details on study area and characteristic of debris flow (as volume involved and run out area and kind of soil involved) should be added. - The applicability or unapplicability of the model to other case study could be further discussed in the section 6 (Conclusion) - Numbering of sub-paragraph is suggested.*

**Response to Reviewer 1**

We appreciate the helpful comments from the reviewer and thank you for your kind words of encouragement. The manuscript has been revised following the suggestions of the reviewer. Point by point responses are listed as follows:

*1. The authors should modify the structure of the manuscript presenting the method before the case study. The desirable outline of manuscript should be Introduction; Model and related sub-paragraph about model input data, variables and outputs; Case study and results of the model; conclusion.*

Response:

This study mainly focused on the debris flow triggering characteristics and occurrence probability. The application of model to the study area is one part of this study. In particular, in the revised version, we have further addressed the debris flow triggering characteristics including rainfall event and rainfall index. However, the structure of the manuscript could not be completely modified to follow the reviewer's suggestions. As such, in the section of *Application of the empirical model*, we have added a *Procedures* section (section 6-1) describing the model's input data, variables and outputs, etc., before the case study (Lines 8-15, Page 20).

*2. Debris fall initiation should be replaced by debris fall triggering (e.g. lines 17 and 22 p.1)*

Response: All instances of "debris flow initiation" have been replaced with "debris flow triggering" in the revised manuscript following the reviewer's suggestions.

*3. Line 19 p.3 the equation should be labelled.*

Response: The equation has been labelled as Eq. (2) (Line 13, Page 7).

*4. The number of RI equation could be cited in Table 1.*

Response: The number of RI equation has been cited in Table 2 (original Table 1) (Line 3, Page 14).

*5. Figure 1 should be improved (blue instead of light blue).*

Response: Figure 1(a) has been improved following the reviewer's suggestion (Page 4).

*6. More details on study area and characteristic of debris flow (as volume involved and run out area and kind of soil involved) should be added.*

Response:

We have added more details of the geological condition in the study area, as shown in Lines 24-28, Page 3; Line 1, Page 4, and Figure 1(b).

The characteristics of significant of debris flows, such as Typhoons Herb, Toraji, and Morakot, have also been collected and presented, as showed in Table 1 (Page 5).

*7. The applicability or unapplicability of the model to other case study could be further discussed in the section 6 (Conclusion) - Numbering of sub-paragraph is suggested.*

Response:

(1) The limitations of the model and the applicability of the model to other cases have been added and addressed in the Conclusion section (No. 5, Pages 23-24).

(2) Sub-paragraphs have been numbered in the Conclusion section.

---

## Author Comment (AC4) · 3 Nov 2017

[revised manuscript text omitted]

rainfalls. Furthermore, the return period (T), also referred to as the recurrence interval, is an important concept that reflects the long-term hydrological characteristics of an area, which is useful for hydrological or hydraulic design. Many studies have analyzed rainfall-triggered shallow landslides using rainfall values obtained by return period (Borga et al., 2002; D'Odorico, 2005; Schilirò et al., 2015; Peres and Cancelliere, 2016; Bogaard and Greco, 2017). However, previous studies mostly focused on hydrological concepts to calculate the return periods of rainfall-triggered shallow landslides. Lack of studies estimated the relationship between the return periods of rainfalls and debris flows occurrence. Therefore, further studies are needed to determine the relationship between the return period T of rainfall characteristics associated with the probability of debris flow occurrence (P) and apply the relationship between P and T to evaluate the probability of debris flow initiation, especially after rainfall and other extreme events.

The Chenyulan watershed in central Taiwan was selected as a study area because it has experienced both the Chi-Chi earthquake and extreme rainfall events. This study had three main purposes: (1) Investigate the debris flow initiation characteristics after extreme rainfall events. Initiation characteristics include the variation in the rainfall index (RI) threshold for debris flow triggering (i.e., the variation in the critical RI), and the recovery period, the time required for the lowered threshold to return to the original threshold. (2) Develop an empirical relationship between P and T for areas affected by extreme rainfalls and the Chi-Chi earthquake. (3) Apply the P–T empirical relationship to evaluate the probability of debris-flow occurrence after extreme rainfall events.

**2. Debris flows in the Chenyulan watershed**

As the study area, the watershed of the Chenyulan River, located in Nantou County, central Taiwan (Figure 1a), has an area of 449 km$^2$, main stream length of 42 km, average stream-bed gradient of 4°, and elevation range of 310–3,952 m. The Chenyulan River follows a major fault, the Chenyulan Fault (Figure 1b), which is a boundary fault dividing two major geological zones of Taiwan. In addition to the boundary fault separating geological zones, the watershed of the Chenyulan River also contains many other faults, accompanied by fractured zones. Consequently, fractured rock mass prevails over the study area, accounting for enormous landslides and providing an abundant source of rock debris for debris

flow (Lin and Jeng, 2000). The annual rainfall in the watershed is between 2,000 and 5,000 mm, with an average of approximately 3,500 mm. Approximately 80% of the annual rainfall occurs in the rainy season between May and October, especially during typhoons, which generally occur three or four times annually. The Chi-Chi earthquake with a moment magnitude $M_W$ 7.6, on September 21, 1999, was the largest to hit Taiwan in 100 years (Shin and Teng, 2001) and it had significant effects on the watershed. In particular, after the Chi-Chi earthquake, the extremely heavy rains brought by Typhoon Toraji in 2001 caused numerous debris flow events in central Taiwan. Owing to the steep topography, loose soils, young (3 million years) and weak geological formations due to the ongoing orogenesis, heavy rainfall, and active earthquakes, many debris flows were triggered by more than 30 rainfall events in past five decades in the watershed (Chen et al., 2013). Notably among these, severe debris flow events were caused by Typhoon Herb in 1996, Typhoon Toraji in 2001, and Typhoon Morakot in 2009. Characteristics for these debris flow events and damages are listed in Table 1.

[Figure]

(a) Location           (b) Geology and lithology

Figure 1: The study area of the Chenyulan watershed in central Taiwan.

Table 1  Severe debris flows events in the Chenyulan watershed

| Date | Trigger | Characteristics for debris flows | Damage |
|---|---|---|---|
| July 31– Aug 01, 1996 | Typhoon Herb | Over 30 significant debris flows occurred in Fengqiu, Tongfu, Dongpu, Shenmu Villages, etc. in Xinyi Township, and Junkeng, Sinshan, Shanan Villages, etc. in Shueili Township, Nantou County. Among these debris flows, the maximum deposited area of debris flow occurred at the gully near Feng-Chiou elementary school with deposited area of 90,900 m$^2$ and the deposited volume is estimated to be as 454,500 m$^3$ (Yu, 1997) | 21 deaths, six injured, over 40 houses destroyed, over 40 ha fruit orchard damaged, significant damage to roads, dams, river regulation works, and other properties. (Lin and Jeng, 2000; Cheng et al., 2005; Jan and Chen, 2005) |
| July 29– 30, 2001 | Typhoon Toraji | Over 70 debris flows occurred in Fengqiu, Tongfu, Shenmu Villages, etc. in Xinyi Township, and Junkeng, Sinshan, Shanan Villages, etc. in Shueili Township, Nantou County. Among these debris flows, the maximum deposited area of debris flow occurred at the San-Bu-Ken gully, Shang-An Village with deposited area of 39,000 m$^2$ and the deposited volume is estimated to be as 195,000 m$^3$ (DPRC, 2001) | 19 deaths, over 30 missing and 70 houses destroyed, significant damages to dikes, roads, bridges and buildings. (Cheng et al., 2005; DPRC, 2001) |
| Aug 06– 11, 2009 | Typhoon Morakot | Over 40 debris flows occurred in Tongfu, Dongpu, Shenmu Villages, etc. in Xinyi Township, Nantou County. (Chen et al., 2011) | Over 20 houses were buried by debris flows or washed away by floods. No death in this event because of many structure and non-structure countermeasures conducted after Typhoon Herb and Typhoon Toraji. Especially, the non-structure countermeasures for the plan of evacuation and shelter and education of hazard prevention. |

**3 Regional rainfall, rainfall index, and extreme rainfall events**

**3-1 Regional average rainfall**

To investigate the variation in rainfall characteristics in Chenyulan watershed, long-term rainfall records were obtained from three meteorological stations (Sun Moon Lake, Yushan, and Alisan stations, as shown in Figure 1a). The rainfall data used in this study are limited to hourly rainfall data because minute-scale rainfall data, such as 5 or 10-min rainfall data were not available. The hourly rainfall data collected from these three stations between 1963 and 2016 were used to estimate the regional rainfall characteristics for the entire Chenyulan watershed, via the reciprocal-distance-squared method (Chow et al., 1988). The estimated point using this method was taken at the centroid of the watershed area. The regional average rainfall in the Chenyulan watershed by the reciprocal-distance-squared method can be expressed as (Chen et al., 2012):

$$I=0.099I_1+0.387I_2+0.514I_3 \tag{1}$$

where $I_1$, $I_2$, and $I_3$, represent the hourly rainfall record from the Sun Moon Lake, Yushan, and Alisan meteorological stations, respectively. The rainfall characteristics estimated by the reciprocal-distance-squared method may not actually reflect the rainfall characteristics at specific locations when local rainfall varied significantly owing to abrupt changes in elevation, but it is a simple method to directly compute the regional average rainfall characteristics for a watershed. Moreover, the regional average rainfall estimated using the reciprocal-distance-squared method can easily represent the variation trend for regional rainfall characteristics throughout Chenyulan watershed and it was used to analyze the characteristics of rainfall triggering of debris flows in the watershed.

**3-2 Identification of rainfall event**

According to the hourly rainfall data of regional average rainfall, the rainfall event can be identified by following a certain criterion. A rainfall event is defined as that when hourly rainfall depth is greater than 4 mm, which is regarded as the beginning of a rainfall event. When hourly rainfall depth remains less than 4 mm continuously for 6 h, the end of that rainfall event is marked. The criterion has been generally used in Taiwan to identify rainfall events for analyzing rainfall events triggering debris flow (Jan et al., 2004). Thirty-eight rainfall events, including 18 rainstorms and 20 typhoon-induced heavy rainfall events, have caused debris flows in the Chenyulan watershed, as listed in Table 2. The maximum hourly rainfall depth $I_m$, the maximum 24-h rainfall amount $R_d$, and the number of debris flows for each rainfall even are also shown in Table 2. The number of debris flows N were collected from Chen et al. (2013), and N was identified through interpretation of aerial photographs, satellite images or/and field investigations. Most rainfall events, accounting for 87%, caused four or less debris flows (N<4 or N=1) and 29% caused one (N=1) in the watershed. Five rainfall events caused ten or more debris flows in the watershed. The five extreme rainfall events included Typhoon Herb (TH) in 1996, Typhoon Toraji (TT) in 2001, Typhoon Mindulle (TMi) in 2004, a heavy rainstorm (HR) in 2006, and Typhoon Morakot (TM) in 2009.

**3-3 Rainfall index**

Rainfall parameters such as peak hourly rainfall, daily rainfall, maximum daily rainfall, cumulative rainfall, average rainfall intensity, and rainfall duration have been used by previous researchers to

investigate the occurrence of debris flows. The choice of rainfall parameters reflects different research objectives and they have been discussed by Chen et al. (2013). Extreme rainfall refers to events of relatively high rainfall intensity and/or high cumulative rainfall. Debris flows caused by a rainfall event generally occurred within the period of the maximum 24-h rainfall $R_d$, and were closely related to the maximum hourly rainfall $I_m$ (Lin and Jeng, 2000; Chen et al., 2011; Chen et al., 2012). $I_m$ and $R_d$ for debris flow events have been used to analyze variations of extreme rainfall events in Chenyulan watershed. However, the occurrence of debris flow for extreme rainfall events is related to not only accumulated rainfall but also rainfall intensity. For using $R_d$ or $I_m$, as shown in Figures 2(a) and 2(b), at $R_d > 580$ mm or $I_m > 54$ mm/h, only five of the eight rainfall events caused 10 or more debris flows. It is inappropriate to apply a single rainfall parameter such as $R_d$ or $I_m$ as a critical condition for the occurrence of multiple debris flows (Chen et al., 2013). Thus, a triggering index RI, an index combining $R_d$ and $I_m$, is proposed and expressed as follows

$$RI = R_d \times I_m \qquad\qquad (2)$$

RI could be used as a critical condition for the occurrence of multiple debris flows, as shown in Figure 2(c). Five extreme rainfall events (TH, TT, TMi, HR, and TM) caused ten or more debris flows in the watershed and they had the highest RI values of all rainfall events from 1963 to 2016, with RI > 365 cm$^2$/h.

[Figure]

(a) $R_d$ index

(b) $I_m$ index

[Figure]

(c) RI index

Figure 2: Variations in three parameters of rainfalls for all rainfall events triggering debris flows between 1963 and 2016 in the Chenyulan watershed (modified from Chen et al., 2013). The Chi-Chi earthquake (CCE) in 1999 had significant effects on the watershed. Five rainfall events, including Typhoon Herb (TH), Typhoon Toraji (TT), Typhoon Mindulle (TMi), a heavy rainstorm (HR), and Typhoon Morakot (TM), caused ten or more debris flows in the watershed.

**3-4 Variations in the rainfall index**

Extreme rainfall events and the Chi-Chi earthquake (CCE) have been shown to affect the critical conditions required for the occurrence of debris flows, and the critical RI values for the occurrence of debris flows have been classified into four categories (Chen et al., 2013): the periods before TH, between TH and CCE, between CCE and TMi, and between TMi and TM (Figure 2). These periods had critical RI values of approximately 165, 60, 2, and 100 $cm^2$/h, respectively. These trends showed that TH caused numerous landslides and debris flows in the watershed, which reduced the critical rainfall threshold for debris flows in subsequent years and the CCE significantly lowered the critical rainfall threshold for debris flow occurrence in the subsequent five years. After the CCE, the critical RI dropped sharply to approximately 2 $cm^2$/h, which was 30 times lower than that before the CCE (critical RI = 60

cm$^2$/h). The results also showed that, approximately five years after the CCE, the critical RI gradually recovered from 2 cm$^2$/h to 100 cm$^2$/h (i.e., the critical RI between TMi and TM).

**4. Variations in the rainfall index after extreme events**

5    The extreme events in the Chenyulan watershed included a severe earthquake, the CCE, and the five extreme rainfall events (TH, TT, TMi, HR, and TM). The extreme rainfall events and the severe earthquake affected the critical condition for debris flow occurrence. Here, the index $r_{RI}$, defined as the ratio of critical RI to the original RI, was used to evaluate the affected period for the variation of RI after an extreme event. The original RI equals 165 cm$^2$/h (i.e., the critical RI before TH), as shown in

10   Figure 2c, and it represent the critical RI unaffected by extreme events such as extreme rainfall events and the CCE. $r_{RI} < 1.0$ indicates that the critical RI to trigger debris flow is lower than that unaffected by extreme events.  $r_{RI} = 1.0$ indicates that the critical RI after an extreme event is equal to that before TH (= 165 cm$^2$/h), and the critical RI has returned to that unaffected by extreme events.

**15   4-1 Critical lines after extreme events**

Table 2 lists numerous debris-flow events triggered by rainstorms and typhoons between 1996 and 2016.  The estimated period t from the time of an extreme event, including TH, CCE, TT, TMi, HR, or TM, and the $r_{RI}$ value between extreme events are also presented. A total of six data ranges for $r_{RI}$ against t are plotted in Figure 3. Most ranges showed that the minimum $r_{RI}$ (the maximum reduction of critical RI) generally occurs at the initial stage after an extreme event and the lower bound of $r_{RI}$ has an

20   increasing tendency over the course of time. The minimum $r_{RI}$ values for events after TMi, HR, TH, TM, TT, and CCE are 0.82, 0.63, 0.37, 0.33, 0.03 and 0.01, respectively.

The critical line of $r_{RI}$ after an extreme event is also presented in Figure 3. The critical lines are empirically determined according to the lower bound of the data range and assume that the line at the

25   initial stage follows the minimum $r_{RI}$ and has a tendency of linear increase. The data ranges after TH and after TM have the same critical line because their lower bounds are close. The required period for the critical RI from the drop down to its original value, referred to as the recovery period herein, also

could be obtained by the critical line at $r_{RI}$ =1.0. The recovery periods are 1.2, 2.1, 3.2, and 4.5 yr for events after TMi, HR, TH and TM, and CCE, respectively.

Among the critical lines, the lowest $r_{RI}$ was caused by the rainfall event after CCE, as shown by the dashed line in Figure 3. The critical rainfall to trigger debris flow after CCE is significantly lowered in the Chenyulan watershed and its affected period could reach five years, approximately in the period between CCE and TMi. The impact of CCE on the critical RI was more significant than those of the other extreme rainfall events. These findings have been discussed and studied by many researches (Lin et al, 2003; Jan and Chen, 2005, Chen, 2011, Chen et al., 2012). Figure 3 also shows that the recovery period affected by CCE was approximately 5 years, in agreement with the results of previous studies. Typhoon Toraji is one of the rainfall events and the only one extreme rainfall within the five years after CCE. Thus, the critical RI line after Typhoon Toraji is much smaller than that caused by other extreme rainfall events because it could be affected by extreme rainfall and CCE.

The possible mechanism for the change in the critical RI threshold (rainfall conditions that triggered debris flow) after an extreme event may be associated with extreme events that caused numerous landslides and/or debris flows in the Chenyulan watershed in the early stage. Landslides and debris flows generally deposit large amounts of loose debris in gullies and on slopes after extreme events (Lin and Jeng, 2000; Lin et al., 2003; Dong et al., 2009; Chen et al., 2012) and increase the volume of loose debris within the watershed. These loose debris generally have lower soil strength and could be located on higher slopes. This results in low pore-water pressure or a low amount of water required to initiate the movement of the soil sediment (Lin et al. 2003, Chen and Jan 2008). Thus, it becomes much easier for debris flow to occur immediately with little rainfall, and that may lead to the lower critical RI or $r_{RI}$ required to trigger debris flow, especially at the early stage after an extreme event. In general, sediments become consolidated and re-orientated with time, the amounts of soil and rock deposited in streams is reduced after each storm, and the shear strength of soil gradually recovers (Fan et al. 2003). In response, this could lead to the gradually increasing critical RI required to trigger debris flow with time after an extreme event.

**4-2 Empirical relationships of RI modification and recovery period**

Excluding the extreme rainfall event of Typhoon Toraji (TT) that could be affected by the Chi-Chi earthquake, four data sets were obtained, either for minimum $r_{RI}$ or for recovery period (Figure 4). These data were obtained from the critical lines in Figure 3 for the period after extreme rainfall events TH, TMi, HR, and TM. The minimum $r_{RI}$ values were 0.82, 0.63, 0.37, and 0.33 for events after TMi (RI=368 $cm^2$/h), HR (RI=529 $cm^2$/h), TH (RI=846 $cm^2$/h), and TM (RI=1078 $cm^2$/h), respectively; and the recovery periods were 1.2, 2.1, 3.2, and 4.5 yr for events after TMi, HR, TH and TM, respectively. Two fit lines for minimum $r_{RI}$ and recovery period against RI are also presented in Figure 4. Minimum $r_{RI}$ decreased and recovery period increased with increasing RI, indicating that after an extreme rainfall event with a higher RI, a lower RI, i.e., a lower value of minimum $r_{RI}$, was required to trigger debris flow and the effect of the extreme rainfall lasted for a longer period. Results of Figure 4 are helpful for modifying the criteria of debris flow warnings. For example, $r_{RI}$ was 0.4 and recovery period was 3.2 years when an extreme rainfall with RI = 900 $cm^2$/h occurred, based on the fit lines showed in Figure 4. This indicates that the critical RI after an extreme rainfall event could be modified to 40% of the original criteria (165 $cm^2$/h) to RI = 66 $cm^2$/h, and the period of critical RI could be lowered to approximately three years.

Table 2: Debris flow events and related rainfall characteristics between extreme rainfalls and the Chi-Chi earthquake in the Chenyulan watershed between 1996 and 2016 (modified from Chen et al., 2013)

| Year | Date of the event | Name of the event | Number of debris flows $N$ | $I_m$ (mm/h) | $R_d$ (mm) | RI (cm$^2$/h) | t (y) | $r_{RI}$ | Analysis range |
|------|------|------|------|------|------|------|------|------|------|
| 1996 | July 31–Aug 01 | Typhoon Herb | 37 | 71.6 | 1181.6 | 846 | | | **TH** |
| 1998 | June 07–08 | Rainstorm | 3 | 28.1 | 227.8 | 64 | 1.85 | 0.39 | |
| 1998 | Aug 04–05 | Typhoon Otto | 4 | 64.6 | 311.7 | 201.4 | 2.01 | 1.22 | **(1)** |
| 1998 | Oct 15–16 | Typhoon Zeb | 2 | 24.6 | 251 | 61.7 | 2.21 | 0.37 | |
| 1999 | May 27–28 | Rainstorm | 2 | 24.3 | 254.3 | 61.8 | 2.83 | 0.37 | |
| 1999 | Sep 21 | Chi-Chi earthquake | | | | | | | **CCE** |
| 2000 | Apr 1 | Rainstorm | 2 | 20 | 75.1 | 15 | 0.53 | 0.09 | |
| 2000 | Apr 25 | Rainstorm | 1 | 8.4 | 30.6 | 2.6 | 0.59 | 0.02 | |
| 2000 | Apr 28–29 | Rainstorm | 1 | 7.9 | 78.2 | 6.2 | 0.61 | 0.04 | |
| 2000 | May 2 | Rainstorm | 1 | 8.1 | 30.6 | 2.5 | 0.61 | 0.02 | |
| 2000 | June 12–14 | Rainstorm | 4 | 18 | 228.1 | 41.1 | 0.73 | 0.25 | |
| 2000 | July 18 | Rainstorm | 3 | 12.7 | 30 | 3.8 | 0.82 | 0.02 | **(2)** |
| 2000 | July 22 | Rainstorm | 3 | 16.3 | 20.7 | 3.4 | 0.84 | 0.02 | |
| 2000 | Aug 5 | Rainstorm | 4 | 11.6 | 38.8 | 4.5 | 0.87 | 0.03 | |
| 2000 | Aug 22–23 | Typhoon Bilis | 2 | 20.6 | 234.5 | 48.3 | 0.92 | 0.29 | |
| 2001 | Jun 5 | Rainstorm | 1 | 7.5 | 27 | 2 | 1.71 | 0.01 | |
| 2001 | June 14–15 | Rainstorm | 3 | 18.4 | 200.1 | 36.8 | 1.73 | 0.22 | |
| 2001 | July 29–30 | Typhoon Toraji | 78 | 78.5 | 587.6 | 461.3 | | | **TT** |
| 2001 | Aug 10 | Rainstorm | 3 | 22.4 | 22.4 | 5 | 0.03 | 0.03 | |
| 2001 | Sep 17 | Typhoon Nari | 4 | 35.7 | 252.5 | 90.1 | 0.13 | 0.55 | |
| 2002 | May 31 | Rainstorm | 4 | 14.4 | 53 | 7.6 | 0.84 | 0.05 | **(3)** |
| 2002 | July 03–04 | Rainstorm | 2 | 13.3 | 117.9 | 15.7 | 0.93 | 0.10 | |
| 2002 | Aug 12 | Rainstorm | 1 | 17.1 | 26.5 | 4.5 | 1.03 | 0.03 | |
| 2004 | July 02–03 | Typhoon Mindulle | 17 | 54 | 681.4 | 368 | | | **TMi** |
| 2004 | Aug 23–25 | Typhoon Aere | 2 | 35 | 385.4 | 134.9 | 0.14 | 0.82 | |
| 2005 | Aug 04–05 | Typhoon Matsa | 1 | 42.3 | 411.9 | 174.2 | 1.09 | 1.06 | **(4)** |
| 2005 | Aug 31–Sep 01 | Rainstorm | 1 | 44.3 | 495 | 219.3 | 1.16 | 1.33 | |
| 2006 | June 08–11 | Heavy Rainstorm | 10 | 77.5 | 682.8 | 529.2 | | | **HR** |
| 2006 | July 13–15 | Typhoon Bilis | 2 | 29.9 | 371.7 | 111.1 | 0.09 | 0.67 | |
| 2007 | Aug 17–20 | Typhoon Sepat | 1 | 31.6 | 328.4 | 103.8 | 1.19 | 0.63 | |

| Year | Date | Event | N | Im | Rd | RI | | | |
|------|------|-------|---|------|--------|--------|------|------|-----|
| 2007 | Oct 06–07 | Typhoon Krosa | 1 | 54.3 | 669.4 | 363.5 | 1.32 | 2.20 | (5) |
| 2008 | July 17–18 | Typhoon Kalmaegi | 3 | 67.2 | 515.7 | 346.6 | 2.10 | 2.10 | |
| 2008 | Sep 12–15 | Typhoon Sinlaku | 2 | 35 | 612.4 | 214.3 | 2.26 | 1.30 | |
| 2009 | Aug 06–11 | Typhoon Morakot | 41 | 85.5 | 1192.6 | 1019.7 | | | TM |
| 2010 | May 23–24 | Rainstorm | 1 | 35.8 | 227.2 | 81.3 | 0.78 | 0.49 | |
| 2011 | July 17–20 | Rainstorm | 1 | 33.6 | 256.2 | 86.1 | 1.94 | 0.52 | |
| 2012 | June 9–12 | Rainstorm | 2 | 33.6 | 384.6 | 129.2 | 2.84 | 0.78 | (6) |
| 2012 | June 18–21 | Typhoon Talim | 1 | 22.2 | 243.4 | 54 | 2.86 | 0.33 | |
| 2012 | Aug 1–3 | Typhoon Saola | 1 | 36.4 | 502.2 | 182.8 | 2.98 | 1.11 | |
| 2013 | July 12–13 | Typhoon Soulik | 3 | 52.4 | 661.7 | 346.7 | 3.92 | 2.10 | |
| ~2016 | No debris flow event | | | | | | | | |

Notes:

1. N = total number of individual debris flows triggered by each rainfall event; Im(mm/h) = maximum hourly rainfall in each rainfall event; Rd(mm) = maximum 24-h rainfall amount in each rainfall event; RI = rainfall index in each rainfall event, and it can be determined by Eq.(2).

2. CCE = Chi-Chi earthquake; TH = Typhoon Herb; TT = Typhoon Toraji; TMi = Typhoon Mindulle; HR = Heavy rainstorm; TM = Typhoon Morakot (TM)

3. (1) = data between TH and CCE; (2) = data between CCE and TT; (3) = data between TT and TMi; (4) = data between TMi and HR; (5) = data between HR and TM; (6) = data after TM.

[Figure]

Figure 3: Relationships between the rainfall index ratio $r_{RI}$ and the period t after extreme rainfalls driven by various values of RI. Four critical lines were determined from the lower bounds of ranges (4), (5), (1) and (6), and (2) in Table 2, representing the critical RI affected by extreme event TMi, HR, TH and TM, and CCE.

[Figure]

Figure 4: Variations of minimum $r_{RI}$ and recovery period after an extreme rainfall driven by the rainfall index RI. Four data sets for minimum $r_{RI}$ or recovery period were obtained from the critical lines in Figure 3 after extreme events TMi (RI=368 cm$^2$/h), HR (RI=529 cm$^2$/h), TH (RI=846 cm$^2$/h), and TM (RI=1078 cm$^2$/h).

**5. Relationship between the probability of debris flow occurrence and return period**

Return period is the average interval in years between events equaling or exceeding a certain magnitude. Return period of rainfall responds the long-term hydrological characteristics of an area and is useful for hydrological or hydraulic design. Therefore, the rainfall index RI associated with return period was determined, and the relationship between the probability of debris flow occurrence and return period was developed.

**5-1 Return period of rainfall**

[revised manuscript text omitted]

**6-1 Procedures**

The empirical model for evaluating the probability of debris flow occurrence was applied through the following steps:

1. Input the hourly rainfall data from three metrological stations and evaluate the regional hourly rainfall using Eq. (1)

2. Determine $I_m$ and $R_d$ from regional rainfall data and calculate the rainfall index RI by Eq. (2)

3. Determine the return period T from Eq. (4b) by giving RI

4. Determine the probability of debris flow occurrence using the P–T relationship.

The P–T relationship of the whole period (WP) is

$$P = \frac{1}{1 + e^{1.11 - 0.98T}}$$

(6)

and the P–T relationship of the extreme rainfall-affected period (ERAP) is

$$P = \frac{1}{1 + e^{3.97 - 3.89T}}$$

(7)

where T is associated with RI and can be determined by Eq. (4), i.e., $T = (RI/180)^{2.27} + 0.98$. The probability of debris flow occurrence (P) can be determined when RI is given according to Eq. (6) or Eq. (7).

However, the two equations were developed based on different periods and different data sets, and the valid conditions for the two equations may not be identical. Eq. (6) predominantly reflects the long-term characteristics of debris flow occurrence, and cannot reflect the short-term characteristic caused by extreme events. In contrast, Eq. (7) focuses on the influence of extreme rainfall events. Hence, field data

of debris flow occurrence and rainfall between 2012 and 2014 were collected to assess the proposed equations.

**6-2 Results**

5      Figure 7 shows the variation in the predicted probability of debris flow occurrence P (blue dot) derived from Eq. (6) for rainfall events with return period greater than one year (T >1 yr) from 2012 to 2014, and labels debris flow events. There were four debris flow events triggered by the rainstorm and Typhoon Talim in 2012, Typhoon Saola and Typhoon Soulik in 2013 (Table 2). Most debris flow events, three of four, reasonably predicted that three predicted P values exceeding 50%. One debris flow event

10     during Typhoon Talim was not predicted successfully in association with the events occurring within three years after the extreme rainfall event of Typhoon Morakot. The RI for debris flow occurrence decreased in the early stage after the extreme rainfall event owing to the fact that extreme rainfalls result in large amounts of loose debris in gullies and on slopes. When the P–T relationship in ERAP (Eq. (7)) was used, instead of Eq. (6), the predicted P (red dot) with P > 50% was in agreement with the field data

15     of debris flow occurrence, as shown in Figure 8.

[Figure]

Figure 7: Application of the probabilistic model of debris flow occurrence in the whole period (WP) between 2012 and 2014.

[Figure]

Figure 8: Probabilistic model of debris flow occurrence in the whole period (WP) compared with that in the rainfall-affected period (ERAP) within the three years after Typhoon Morakot. There are three debris flow occurrence events. Three rainfall events show the predicted probability of debris flow occurrence P exceeding 50% by the model in ERAP, and two rainfall events show the predicted P exceeding 50% by the model in WP.

**7. Conclusions**

1.  Debris flows and their corresponding rainfall events in the Chenyulan watershed, central Taiwan were investigated. The rainfall index RI, defined as the product of $I_m$ and $R_d$, was used to analyze the rainfall conditions critical for debris flow occurrence after extreme events. The extreme events included the Chi-Chi earthquake in 1999 and five extreme rainfalls in 1996, 2001, 2004, 2006, and 2009. The extreme rainfall events and the Chi-Chi earthquake affected the critical condition for the occurrence of debris flows. The rainfall index RI could reflect the debris flow initiation characteristics after extreme rainfall events.

2.  The RI threshold for the occurrence of debris flows was reduced in the years following an extreme rainfall event. Reduced RI values showed an increasing trend over time, and it gradually returned to the original RI, representative of the RI unaffected by the extreme rainfall. The required time, i.e., the recovery period, for the decreased RI to return to the original value for

extreme rainfalls was analyzed. The reduction in RI as well as the recovery period were influenced by the RI. The RI at the early stage after an extreme rainfall showed the maximum decrease of approximately 30% of the original RI. The maximum the recovery period was approximately three years. Understanding the reduced RI and the recovery period is helpful for modifying the criteria of debris flow warnings.

3.  The rainfall index RI associated with return period was analyzed. The extreme events triggering numerous debris flows, excluding events affected by the Chi-Chi earthquake (CCE), mostly had return period exceeding 10 years. The return period for the critical RI affected by the CCE was approximately 1 year, much smaller than that affected by extreme events. The empirical relationships between the probability of debris flow occurrence P and return period based on the Chi-Chi earthquake and extreme rainfalls were developed. P increased significantly after extreme rainfall events or the Chi-Chi earthquake at the same return period. In particular, the P value influenced by the Chi-Chi earthquake was markedly higher than that affected by extreme rainfall events.

4.  The relationship between the probability of debris flow occurrence P and the return period T was applied to evaluate P during recent rainfall events after the extreme rainfall of Typhoon Morakot, which showed that the model was reasonable for explaining debris flow occurrence. The benefits of developing the P–T relationship include that P values can be evaluated at various T values (or different rainfall conditions) to understand how the probability of debris flow occurrence is affected by the Chi-Chi earthquake or extreme rainfall.

5.  The empirical model for evaluating the probability of debris flow occurrence was developed based on the regional characteristics of the Chenyulan watershed and it may not be applicable to areas with different hydrogeological properties. The reasonability of this model must be assessed and empirical coefficients are required for calibration if the model is to be applied to other area. In addition, this model mainly reflects the overall critical rainfall conditions to trigger debris flow in a region after extreme events, and it may not be valid for the evaluation of the probability of debris flow occurrence for small catchments with single debris flow. To verify the applicability of

the model to other case studies, more detailed data must be collected and analyzed in order to establish empirical formulas and models.

**Appendix A**

List of symbols and abbreviations

| | |
|---|---|
| CCE | Chi-Chi earthquake |
| CCEAP | Chi-Chi earthquake-affected period |
| ERAP | Extreme rainfall-affected period |
| HR | Heavy rainstorm in 2006 |
| I | Region hourly rainfall |
| $I_1, I_2, I_3$ | Hourly rainfall record from the Sun Moon Lake, Yushan, and Alisan meteorological stations, respectively. |
| $I_m$ | Maximum hourly rainfall during each rainfall event |
| N | Total number of individual debris flows triggered by each rainfall event |
| $N_D$ | Number of rainfall events that have triggered debris flows |
| $N_R$ | Number of rainfall events |
| n | Number of years in the record |
| m | Rank of a value in a list ordered by descending magnitude |
| P | Probability of debris flow occurrence |
| $R_d$ | Maximum 24-h rainfall amount during each rainfall event |
| RI | Rainfall index, $RI = R_d I_m$ |
| $r_{RI}$ | Ratio of RI to original RI (=165 $cm^2$/h) |
| T | Return period of rainfall |
| TH | Typhoon Herb |
| TM | Typhoon Morakot |
| TMi | Typhoon Mindulle |
| TT | Typhoon Toraji |
| t | Period after an extreme rainfall event, the estimated period from the time of an |

| | extreme event |
| --- | --- |
| WP | Whole period, 1985–2016 |
| $\alpha$ and $\beta$ | Empirical coefficients in Eq. (5) |

**Acknowledgement**

This study received financial support from the Ministry of Science and Technology, Taiwan (MOST 105-2625-M-211-001).

---

## Author Comment (AC5) · 3 Nov 2017

please see the attached file.

Please also note the supplement to this comment:
https://www.nat-hazards-earth-syst-sci-discuss.net/nhess-2017-265/nhess-2017-265-AC5-supplement.pdf
* * *